# Heavy Tau Burden with Subtle Amyloid β Accumulation in the Cerebral Cortex and Cerebellum in a Case of Familial Alzheimer’s Disease with APP Osaka Mutation

**DOI:** 10.3390/ijms21124443

**Published:** 2020-06-22

**Authors:** Hiroyuki Shimada, Shinobu Minatani, Jun Takeuchi, Akitoshi Takeda, Joji Kawabe, Yasuhiro Wada, Aya Mawatari, Yasuyoshi Watanabe, Hitoshi Shimada, Makoto Higuchi, Tetsuya Suhara, Takami Tomiyama, Yoshiaki Itoh

**Affiliations:** 1Department of Radiology, Osaka City University Graduate School of Medicine, Osaka 545-8585, Japan; h.shimada@med.osaka-cu.ac.jp; 2Department of Neurology, Osaka City University Graduate School of Medicine, Osaka 545-8585, Japan; m4518413@med.osaka-cu.ac.jp (S.M.); jun-t@med.osaka-cu.ac.jp (J.T.); a-taked@med.osaka-cu.ac.jp (A.T.); 3Department of Nuclear Medicine, Osaka City University Graduate School of Medicine, Osaka 545-8585, Japan; kawabe@med.osaka-cu.ac.jp; 4RIKEN Center for Biosystems Dynamics Research, Kobe 650-0047, Japan; yasuwada@riken.jp (Y.W.); a.mawatari@riken.jp (A.M.); yywata@riken.jp (Y.W.); 5Department of Functional Brain Imaging Research (DOFI), National Institutes for Quantum and Radiological Science and Technology (QST), Chiba 263-8555, Japan; shimada.hitoshi@qst.go.jp (H.S.); higuchi.makoto@qst.go.jp (M.H.); suhara.tetsuya@qst.go.jp (T.S.); 6Department of Translational Neuroscience, Osaka City University Graduate School of Medicine, Osaka 545-8585, Japan; tomi@med.osaka-cu.ac.jp

**Keywords:** amyloid PET, tau PET, amyloid precursor protein, mutation, familial Alzheimer’s disease, cerebellum, PBB3, PiB

## Abstract

We previously identified a novel mutation in amyloid precursor protein from a Japanese pedigree of familial Alzheimer’s disease, FAD (Osaka). Our previous positron emission tomography (PET) study revealed that amyloid β (Aβ) accumulation was negligible in two sister cases of this pedigree, indicating a possibility that this mutation induces dementia without forming senile plaques. To further explore the relationship between Aβ, tau and neurodegeneration, we performed tau and Aβ PET imaging in the proband of FAD (Osaka) and in patients with sporadic Alzheimer’s disease (SAD) and healthy controls (HCs). The FAD (Osaka) patient showed higher uptake of tau PET tracer in the frontal, lateral temporal, and parietal cortices, posterior cingulate gyrus and precuneus than the HCs (>2.5 SD) and in the lateral temporal and parietal cortices than the SAD patients (>2 SD). Most noticeably, heavy tau tracer accumulation in the cerebellum was found only in the FAD (Osaka) patient. Scatter plot analysis of the two tracers revealed that FAD (Osaka) exhibits a distinguishing pattern with a heavy tau burden and subtle Aβ accumulation in the cerebral cortex and cerebellum. These observations support our hypothesis that Aβ can induce tau accumulation and neuronal degeneration without forming senile plaques.

## 1. Introduction

Senile plaques and neurofibrillary tangles (NFTs) in the brain are hallmarks of Alzheimer’s disease (AD). These pathological changes can be visualized and assessed clinically by positron emission tomography (PET) with radioisotope-labeled probes specific for fibrillar Aβ and tau, such as ^11^C-Pittsburgh compound-B (PiB) for amyloid and ^11^C-pyridinyl-butadienyl-benzothiazole 3 (PBB3) for NFTs [1,2,3]. Since the existence of senile plaques is a prerequisite for the pathological diagnosis of AD, only individuals with dementia who are shown to be positive for amyloid by PET fulfill the recent clinical criteria for typical AD, including the IWG-2 criteria [4]. However, the view that the true culprit that initiates AD is not senile plaques but pathologically invisible, small oligomeric aggregates of Aβ has been widely accepted [5,6]. Animal and organotypic experiments have suggested that Aβ oligomers cause the synaptic and cognitive dysfunction as well as early pathological changes in AD, including tau hyperphosphorylation [7,8,9]. However, in humans, it is still unclear whether AD develops only with Aβ oligomers, or whether Aβ oligomers can induce the later pathologies of AD, including NFTs, in the absence of senile plaques.

Previously, we identified a novel mutation in amyloid precursor protein (APP) from a pedigree of familial AD in Osaka, Japan, FAD (Osaka) [10]. This ‘Osaka’ mutation is the deletion of codon 693 of the APP gene, resulting in mutant Aβ that lacks the 22nd glutamate. Only homozygous carriers suffer from dementia, indicating that this mutation is recessive. In vitro studies revealed that this mutation has a very unique characteristic that accelerates Aβ oligomerization but does not form amyloid fibrils [10]. Transgenic (Tg) mice expressing human APP with this mutation (APP_OSK_ mice) displayed intraneuronal accumulation of Aβ oligomers followed by synaptic and cognitive impairment, tau hyperphosphorylation, glial activation, and neurodegeneration but not amyloid plaques [11]. Furthermore, double Tg mice expressing both APP_OSK_ and wild-type human tau demonstrated that NFTs are also induced by Aβ oligomers alone [12].

To confirm these findings in humans, we evaluated amyloid accumulation in two sister patients harboring FAD (Osaka) [13]. PiB-PET scans revealed almost negligible amounts of Aβ accumulation in both patients, supporting our speculation that this mutation causes disease without forming senile plaques. To further explore the relationship between Aβ, tau and neurodegeneration, we performed tau and Aβ imaging in the proband of FAD (Osaka) and in patients with sporadic AD (SAD) and healthy controls (HCs) using PET and MRI.

## 2. Results

### 2.1. Demographic Data

Table 1 shows the demographic data for the groups that included the FAD (Osaka) patient (*n* = 1), early sporadic AD (early SAD) patients (*n* = 6), an advanced SAD patient (*n* = 1) and healthy controls (HCs) (*n* = 12). The age of the FAD (Osaka) patient at the time of the imaging, 70 years old, was comparable to that of the early SAD patients (mean ± standard deviation (SD), 69.7 ± 12.4 years old). The HCs were chosen to match the age distribution of the patients with AD (71.8 ± 8.7 years old).

In contrast, the disease duration of the FAD (Osaka) patient at the time of tau imaging, 14 years, was much longer than that of the early SAD patients (3.1 ± 1.7 years) and, to a lesser extent, longer than that of the advanced SAD patient (6 years). At this time, the FAD (Osaka) patient was in an extremely advanced stage with a Mini-Mental State Examination (MMSE) score of 0 points, whereas the early SAD patients had MMSE scores just below the cutoff level for dementia. One patient with SAD was in an advanced stage and unable to communicate at all. Subjects in the HC group lost few or no points on the cognitive test.

### 2.2. MRI Study

The FAD (Osaka) patient had severely advanced brain atrophy, including most of the cerebral cortex and brain stem shown in a T1 weighted MRI (Figure 1). Parahippocampal atrophy and ventricular enlargement were prominent in the coronal section. The cerebellum and primary motor cortex were relatively spared. These changes in FAD (Osaka) were all noticeable by comparing FAD (Osaka) to HC (Figure 1).

In contrast, representative images of the early SAD group showed only minor hippocampal atrophy with no other cortical involvement on MRI. The patient with advanced SAD had diffuse cortical atrophy with ventricular enlargement.

### 2.3. Tau PET Imaging

Representative tau PET images of the 4 groups are shown in Figure 2. The standard uptake value ratio (SUVR) values of accumulated PBB3 with reference to the midbrain were calculated. In the FAD (Osaka) patient, increased retention of ^11^C-PBB3 was noticeable in most of the cerebral cortex except for the medial temporal cortex, including the hippocampus. The accumulation was also prominent in the cerebellar cortex. In contrast, elevated ^11^C-PBB3 radio signals were small and limited to the frontal, parietal, and lateral temporal cortex, and the precuneus and the posterior cingulate gyrus in the early SAD patients. Furthermore, the advanced SAD patient showed diffusely increased ^11^C-PBB3 signals in the cerebral cortex, but they seemed less increased than that of the FAD (Osaka) patient. No increase in the ^11^C-PBB3 signals was found in HC.

The regional SUVR values with reference to the midbrain in each group are shown in Figure 3. In the cerebral cortex, ^11^C-PBB3 accumulation in the FAD (Osaka) patient was higher than that in the HCs in all measured regions, including the frontal cortex, lateral temporal cortex, posterior cingulate gyrus, precuneus and parietal cortex (>2.5 SD). ^11^C-PBB3 accumulation was also higher than that of both the early and advanced SAD groups, especially in the lateral temporal and parietal cortex (>2 SD and >2.5 SD, respectively, in early SAD). In contrast, remarkable uptake of ^11^C-PBB3 in the cerebellum was found only in the FAD (Osaka) patient.

### 2.4. Amyloid β PET Imaging

Figure 4 shows representative Aβ PET images taken with ^11^C-PiB in the 4 groups. The SUVR values of accumulated PiB with reference to the cerebellum were calculated and projected on the MRI images of each patient. In the FAD (Osaka) patient, the accumulation of ^11^C-PiB was negligible in most of the cerebral cortex except for the temporal cortex and limited parts of the parietal and frontal cortices where very low accumulation could be seen. Relatively elevated ^11^C-PiB uptake was found in the cerebellar cortex compared to the cerebral cortex, and the more severe atrophy in the cerebral cortex than in the cerebellum might affect the apparent ^11^C-PiB accumulation.

In contrast, elevated ^11^C-PiB uptake was remarkable in the frontal, parietal and lateral cortices in both the early and advanced SAD patients. The HC patients showed no evident increase in ^11^C-PiB accumulation.

According to the J-ADNI PET core criteria [14], PiB uptake is regarded as positive when the cortical accumulation is higher than that in the white matter just below the cortex, as shown in the SAD patients in Figure 4. The negative patient had a reversed pattern of PiB, as in the HC patients. In the FAD (Osaka) patient, PiB appeared to be higher in the cortex compared to the white matter in the temporal, parietal and frontal cortices, but the severe cortical atrophy made it quite difficult to discern.

### 2.5. Scatter Plot Analysis of Tau vs. Aβ PET

Scatter plot analysis of the two tracers revealed that the FAD (Osaka) patient exhibited a distinguishing pattern with a high tau burden and subtle Aβ accumulation in all parts of the AD signature cortices (Figure 5). In contrast, the SAD patients exhibited high accumulation of both tau and Aβ, whereas both were negligible in the HCs.

## 3. Discussion

This report is the first report of tau PET imaging in the proband of familial AD with Osaka mutation, which revealed heavy tau burden in the cerebral cortex and cerebellum with only negligible Aβ in the cerebral cortex. The findings indicate that Aβ may induce tau accumulation and neuronal degeneration without forming senile plaques as previously reported in basic experiments [10,11,12].

We previously reported that in SAD, tau accumulation spreads from the parahippocampal gyrus to the cerebral cortex with advancing phases of AD, whereas Aβ distribution is already advanced in the clinically earliest stage [15]. Similar reports supporting our data have been recently published by Jack et al. [16]. The increase in tau accumulation in a patient with advanced SAD compared with early SAD patients was also confirmed in the present study. The FAD (Osaka) patient also exhibited higher tau accumulation than the HCs in all the AD signature ROIs in the cerebral cortex. It is noteworthy that the accumulation was even higher than early SAD by more than 2 SD in the lateral temporal cortex and parietal cortex. Although the difference between FAD (Osaka) and advanced SAD was also noticeable in these ROIs, whether this distribution pattern is specific to FAD (Osaka) or just reflecting the advanced stage of AD in general, requires further validation with more advanced SAD cases.

As is widely known, Aβ accumulates heavily and diffusely in the cerebral cortex even in the early stages of SAD and remains high in the advanced stage as in the present study. In contrast, Aβ accumulation was negligibly low in earlier cases of FAD (Osaka) previously reported [1] and remained low in the advanced stage in the present study. These findings strongly support our hypothesis that accumulated Aβ that is usually found with senile plaques is not a prerequisite for the disease progression that includes tau accumulation and neurodegeneration in any phase of FAD (Osaka). Instead, small oligomeric aggregates of Aβ, not detectable with PiB, may play a key role in the onset and progression of the disease. These mechanisms were experimentally confirmed by our previous studies revealing that the Osaka mutation promotes Aβ oligomerization in the brain without forming senile plaques [10,11,12]. In addition to Aβ, more recent evidences have pointed to the pertinent role of soluble oligomeric tau in AD onset and progression. These oligomers may share a common mechanism of toxicity [17]. As the present case of FAD (Osaka) exhibited heavy tau burden on PBB3 PET, any other FAD mutation without tau accumulation, if found, might prove the toxicity of tau oligomer.

The possible non-harmful profile of accumulated Aβ is critical for the development of AD treatment. Previous clinical trials targeting accumulated Aβ were mostly successful in reducing Aβ but failed to improve the clinical outcomes and even worsened the outcomes in some studies [18]. In contrast, in the present and previous studies, we demonstrated that tau accumulates more widely as the disease progresses in SAD [15]. The burden of tau was also heavy in FAD (Osaka) without amyloid accumulation. These findings strongly warrant further development of novel treatments to reduce soluble Aβ oligomer or target tau, including aducanumab, which is a human monoclonal antibody that selectively binds to Aβ fibrils and soluble oligomers [19].

Cerebellar lesions with senile plaques and/or neurofibrillary tangles as well as Purkinje cell loss were pathologically reported in some FAD patients [20,21,22,23], whereas these pathological changes were usually not found until the end stage in SAD [24]. In the present study, Aβ and tau accumulation in the cerebellum were only found in FAD (Osaka). The present patient and other family members with dementia symptoms demonstrated cerebellar ataxia even in the early phase, which is quite rare in most patients with SAD. Cerebellar accumulation of Aβ or tau may be helpful in distinguishing FAD from SAD if familial history cannot be obtained.

In the advanced stage of AD, neuronal degeneration results in cerebral atrophy. The process may decompose and reduce the actual accumulation of Aβ and tau. In addition, atrophy reduces apparent tracer accumulation. The accumulation of PBB3 was even higher in the lateral temporal and parietal cortex in the FAD (Osaka) patient than in the advanced SAD patients with similar cerebral atrophy, indicating substantial tau accumulation in these regions. In contrast, PiB accumulation seemed much lower in the patient with FAD (Osaka) than in the patient with advanced SAD, suggesting much lower Aβ accumulation in FAD (Osaka), but the small amount of Aβ in the FAD (Osaka) patient compared to that in the HCs cannot be ruled out completely.

## 4. Materials and Methods

### 4.1. Ethical Approval

All procedures performed in studies involving human participants were approved by the Institutional Research Ethics Committee of Osaka City University Graduate School of Medicine (IRB# 3009 approved on 25 December 2014) and were conducted in accordance with the 1964 Helsinki Declaration and its later amendments.

### 4.2. Informed Consent

Written informed consent was obtained from all participants or from close family members when the participants were cognitively impaired.

### 4.3. Subjects

#### 4.3.1. Familial Alzheimer’s Disease with APP Mutation

Details of the FAD (Osaka) pedigree were described elsewhere [10]. We previously reported PET imaging of Aβ accumulation in two sister patients of the pedigree [13]. In the present study, the elder sister, who was the proband, was examined. She was in the advanced stages of dementia, bed-ridden and did not speak.

#### 4.3.2. Sporadic AD

We recruited SAD patients from patients attending the memory clinic of Osaka City University Hospital. We evaluated medical history, neurological findings and general blood tests. Two qualified clinical psychologists (M.A. and N.K.) scored the MMSE. An MRI of the brain including coronal sections for hippocampal evaluation was taken.

AD was diagnosed based on the IWG-2 criteria for typical AD with amyloid PET as in vivo evidence of AD pathology [4]. Apparent familial AD was not included in the SAD diagnosis. The same exclusion criteria applied to the HCs were applied to the AD patients to exclude confounding diseases, such as diabetes, dyslipidemia and hypertension.

#### 4.3.3. Healthy Controls

HCs without a history of brain disorders or subjective abnormalities were openly recruited as candidates. All candidates took physical examinations covering vital and cardiopulmonary systems. Tests that were administered to the AD patients were applied to the HCs.

The exclusion criteria were (1) a history of any brain diseases, brain surgery, or head trauma that requires hospitalization; (2) high risk for cerebrovascular diseases, including poorly controlled diabetes, severe dyslipidemia, and hypertension above the recommended level of standard guidelines; (3) any neurological findings suggesting brain diseases; (4) low cognitive test scores [MMSE, and Rivermead Behavioral Memory Test (RBMT)]; (5) significant MRI lesions including asymptomatic lacunas (less than 15 mm in diameter, high in T2, FLAIR and low in T1), severe white matter lesions (more than 1 grade in Fazekas score [25]), more than 4 cerebral microbleeds, and atrophy beyond average for their age by visual inspection; and (6) positive amyloid PET imaging. No other biomarkers were evaluated in the HC group.

### 4.4. PET Data Acquisition

^11^C-PBB3 and ^11^C-Pittsburgh compound-B (2-[4-(^11^C-methylamino) phenyl]-1,3-benzothiazol-6-ol, ^11^C-PiB) were produced following the methods previously reported [2,26,27]. ^11^C-PBB3-and ^11^C-PiB-PET images were obtained with a Siemens Biograph16 scanner (Siemens/CTI, Knoxville, TN, USA) and with an Eminence-B PET scanner (Shimadzu Co., Kyoto, Japan), respectively. ^11^C-PBB3, a tau tracer, was intravenously injected in the range of 370 MBq (body weight ≤ 50 kg) to 555 MBq (body weight ≥ 70 kg) in a dimly lit room to avoid photoracemization. A 60-min PET scan was performed in list mode. The acquired data were sorted into dynamic data with 6 × 10 s, 3 × 20 s, 6 × 60 s, 4 × 180 s, and 8 × 300 s frames. To evaluate Aβ accumulation, each subject received 400 to 500 MBq of ^11^C-PiB intravenously over 1 min. After the injection, a static scan image acquisition was performed for 50 to 70 min. Reconstruction of PET images for ^11^C-PBB3 and ^11^C-PiB was performed by filtered back projection using a 4-mm full width at half maximum (FWHM) Hanning filter and a 5-mm FWHM Gaussian filter with attenuation and scatter correction, respectively.

### 4.5. Criteria for Aβ Accumulation in SAD

Aβ accumulation was determined visually based on the Japanese Alzheimer’s Disease Neuroimaging Initiative (J-ADNI) Visual Criteria for ^11^C-PiB-PET (J-ADNI_PETQC_Ver1.1) modified from the ADNI PET core criteria [28]. The four regions selected for the assessment were the frontal lobe, lateral temporal lobe, lateral parietal lobe and the combined area of precuneus and posterior cingulate gyrus. The patient was considered positive when the accumulation in one of the four cerebral cortices was higher than that in the white matter just below the cortex and negative when none of the four cortices had higher accumulation than the white matter.

### 4.6. MRI Acquisition

We used a 1.5- or 3-Tesla magnetic resonance scanner (MAGNETOM Avanto, Siemens Healthcare, Erlangen, Germany or Ingenia, Philips Healthcare, Best, The Netherlands) with three-dimensional volumetric acquisition of a T1-weighted gradient echo sequence (repetition time range/echo time range, 6.5 ms/3.2 ms; field of view [frequency × phase], 240 mm × 240 mm; matrix, 256 × 256; contiguous axial slices of 1.5 mm thickness).

### 4.7. Image Processing

For all image processes, we used PMOD software version 3.7 (PMOD Technologies Ltd., Zurich, Switzerland) [29]. Acquired ^11^C-PBB3 and ^11^C-PiB images were transformed into standard brain and then SUVR images were reconstructed with the midbrain as the reference region using the frame summation of dynamics of the image for 30 to 50 min after ^11^C-PBB3 injection and for 50 to 70 min after ^11^C-PiB injection. The SUVR level of each volume of interest (VOI) was calculated in a manually set region of interest (ROI) as shown in Appendix A.

### 4.8. Statistical Analysis

To quantitatively evaluate tau accumulation, SUVR values relative to the midbrain were calculated in each ROI. Statistically, the value of FAD (Osaka) was regarded as significantly high when it was higher than the average by more than 2 SD (standard deviation) in SAD patients and HCs.

## 5. Conclusions

Here, we report a patient with familial AD who had heavy tau burden in the cerebral cortex and cerebellum with only a negligible amount of Aβ. The unprecedented observations support our hypothesis that Aβ, probably in oligomers, induces tau accumulation and neuronal degeneration without forming senile plaques.

## Figures and Tables

**Figure 1 ijms-21-04443-f001:**
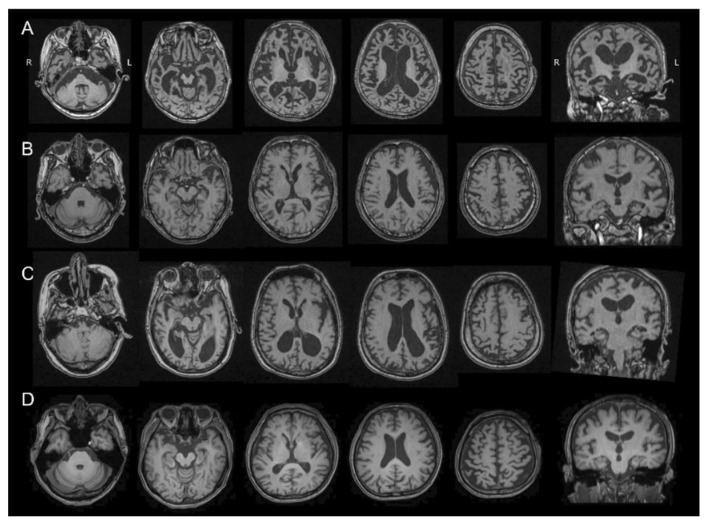
T1-weighted MRI scans of a patient with familial Alzheimer’s disease with Osaka mutation (FAD (Osaka)) (**A**); a patient with early stage sporadic Alzheimer’s disease (early SAD) (**B**); a patient with advanced stage of SAD (**C**); and a healthy control (HC) (**D**). The FAD (Osaka) patient had severely advanced brain atrophy including most of the cerebral cortex and brain stem. Parahippocampal atrophy and ventricular enlargement were prominent in the coronal section. The cerebellum and primary mortar cortex were relatively spared. R: right, L: left.

**Figure 2 ijms-21-04443-f002:**
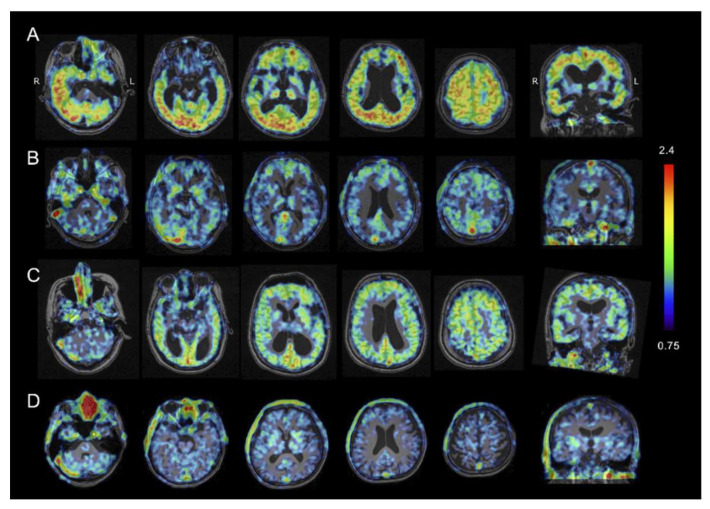
Tau PET using PBB3 in a patient with familial Alzheimer’s disease with Osaka mutation (FAD (Osaka)) (**A**); a patient of early stage of sporadic Alzheimer’s disease (early SAD) (**B**); a patient with advanced stage of SAD (**C**); and a healthy control (HC) (**D**). The heat map range (colored bar) of tau tracer uptake indicates standard uptake value ratio (SUVR) with reference to the midbrain. In the FAD (Osaka) patient, noticeable PBB3 accumulation was observed in the cerebral cortex and the cerebellar cortex, whereas the AD patient had much less tau accumulation that was more localized in the frontal, parietal, and lateral temporal cortices.

**Figure 3 ijms-21-04443-f003:**
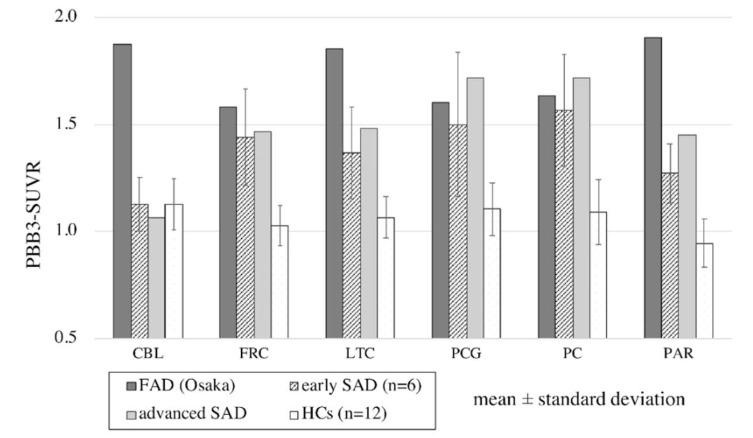
Regional PBB3 uptake with reference to the midbrain in familial Alzheimer’s disease with Osaka mutation (FAD (Osaka)), in early and advanced stage patients with sporadic Alzheimer’s disease (SAD) and healthy controls (HCs). Regions were set in the cerebellum (CBL), frontal cortex (FRC), lateral temporal cortex (LTC), posterior cingulate gyrus (PCG), precuneus (PC) and parietal cortex (PAR). In the cerebral cortex, PBB3 uptake in the FAD (Osaka) patient was higher than that in the HCs in all regions (>2.5 SD). PBB3 uptake was even higher than that in the early stage SAD patients and advanced stage SAD patients in the lateral temporal (>2 SD) and parietal cortices (>2.5 SD). Remarkably elevated PBB3 uptake in the cerebellum was found only in the FAD (Osaka) patient.

**Figure 4 ijms-21-04443-f004:**
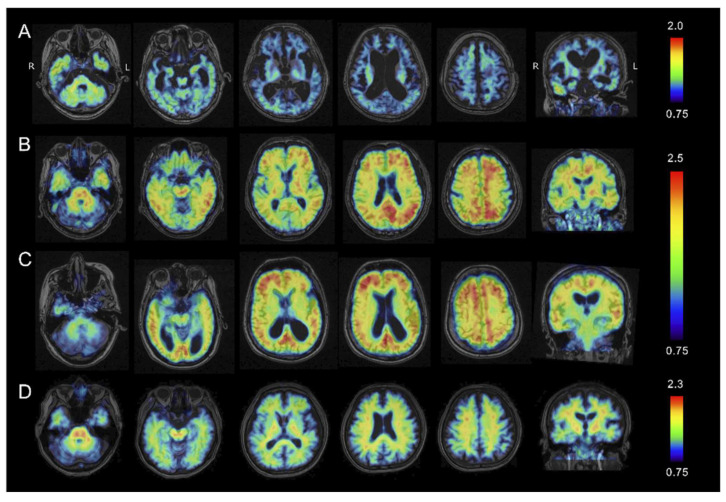
Amyloid PET using PiB in a patient with familial Alzheimer’s disease with Osaka mutation (FAD (Osaka)) (**A**); a patient with early sporadic Alzheimer’s disease (SAD) (**B**); a patient with advanced stage SAD (**C**); and a healthy control (HC) (**D**). Heat map range (colored bar) of amyloid tracer uptake defined by standard uptake value ratio (SUVR) with reference in the cerebellum. In the FAD (Osaka) patient, only negligible amounts of PiB retention were detected in any part of the cerebral cortex. Both the early and advanced SAD patients had highly elevated PiB uptake in the frontal, parietal and lateral cortices. No elevation in PiB accumulation was found in the HC.

**Figure 5 ijms-21-04443-f005:**
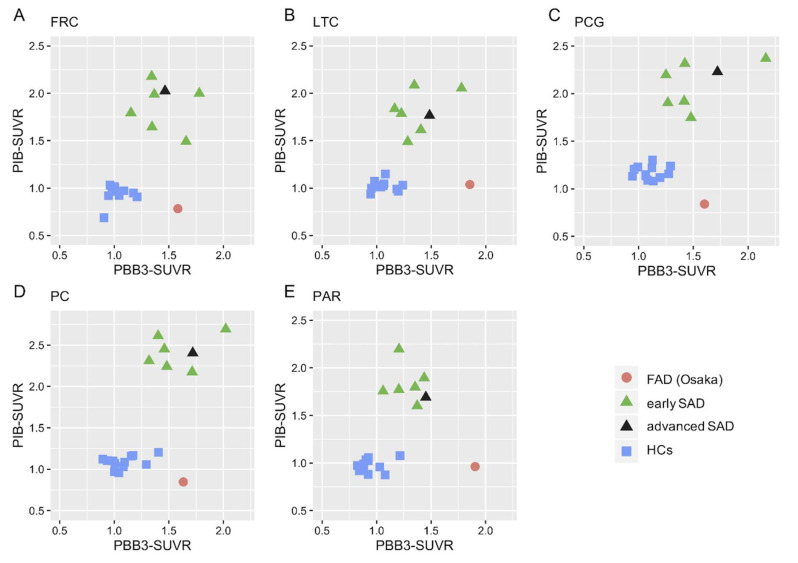
Scatter plot analysis between tau (PBB3-SUVR) and Aβ (PiB-SUVR) accumulation in the frontal cortex ((**A**): FRC); lateral temporal cortex ((**B**): LTC); posterior cingulate gyrus ((**C**): PCG); precuneus ((**D**): PC); and parietal cortex ((**E**): PAR) in a patient with familial Alzheimer’s disease with Osaka mutation (FAD (Osaka)), patients with early and advanced sporadic Alzheimer’s disease (SAD) and healthy controls (HCs). The FAD (Osaka) patient showed a distinguishing pattern with highly elevated PBB3 uptake and subtle uptake of PiB in all regions. In contrast, the SAD patients exhibited highly elevated accumulation of both PBB3 and PiB, whereas both were negligible in HCs.

**Table 1 ijms-21-04443-t001:** Demographic Data.

	*n*	Age	Gender (M/F)	Disease Duration (y)	MMSE
FAD (Osaka)	1	70	0/1	14	0
early SAD	6	69.7 ± 12.4	4/2	3.1 ± 1.7	23.3 ± 3.7
advanced SAD	1	53	0/1	6	0
HCs	12	71.8 ± 8.7	7/5	n.a.	28.8 ± 1.3

FAD (Osaka): familial Alzheimer’s disease in Osaka; SAD: sporadic Alzheimer’s disease; HCs: healthy controls; MMSE: Mini-Mental State Examination; average ± standard deviation; n.a.: not available.

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
