# Peer review of "Heavy Tau Burden with Subtle Amyloid β Accumulation in the Cerebral Cortex and Cerebellum in a Case of Familial Alzheimer’s Disease with APP Osaka Mutation"

_ijms, 2020, doi:10.3390/ijms21124443_

Round 1
Reviewer 1 Report
The authors describe a novel mutation in the amyloid precursor protein from a Japanese (Osaka) familial Alzheimer’s disease (FAD) that exhibits a heavy tau burden and subtle Aβ accumulation in the cerebral cortex and cerebellum, supporting the notion that Aβ can induce tau accumulation and neuronal degeneration without forming senile plaques. FAD (Osaka) mutation results in the lack of the 22nd glutamate, which accelerates Aβ oligomerization but does not form amyloid fibrils or amyloid plaques.
Using PET and MRI the authors demonstrate negligible amounts of Aβ accumulation in patients harboring the FAD (Osaka) mutation, supporting the idea that senile plaques are not necessaries to initiate the disease, and even more important, that small amounts of Aβ oligomers may induce significant tau accumulation and neuronal degeneration.
These results agree with the view that AD initiates with oligomeric aggregates of Aβ inducing the later pathologies in the absence of senile plaques. Interestingly, double transgenic mice expressing both FAD (Osaka) human mutation and wild-type human tau demonstrated that neurofibrillary tangles are induced by Aβ oligomers alone. Do the authors have any biochemical explanation about why the FAD (Osaka) mutation causes oligomeric aggregates that not evolve to fibrils or senile plaques but results in significant tau aggregation?
The article presents significant results to know the role played by mutations within the Aβ sequence and, therefore, may help to understand the amyloidogenic process in AD and its role in the pathophysiology and progression to dementia.
Author Response
(Reviewer) The authors describe a novel mutation in the amyloid precursor protein from a Japanese (Osaka) familial Alzheimer’s disease (FAD) that exhibits a heavy tau burden and subtle Aβ accumulation in the cerebral cortex and cerebellum, supporting the notion that Aβ can induce tau accumulation and neuronal degeneration without forming senile plaques. FAD (Osaka) mutation results in the lack of the 22nd glutamate, which accelerates Aβ oligomerization but does not form amyloid fibrils or amyloid plaques.
(Response) The summary by the reviewer #1 is quite to the point.
(Reviewer) Using PET and MRI the authors demonstrate negligible amounts of Aβ accumulation in patients harboring the FAD (Osaka) mutation, supporting the idea that senile plaques are not necessaries to initiate the disease, and even more important, that small amounts of Aβ oligomers may induce significant tau accumulation and neuronal degeneration.
These results agree with the view that AD initiates with oligomeric aggregates of Aβ inducing the later pathologies in the absence of senile plaques. Interestingly, double transgenic mice expressing both FAD (Osaka) human mutation and wild-type human tau demonstrated that neurofibrillary tangles are induced by Aβ oligomers alone. Do the authors have any biochemical explanation about why the FAD (Osaka) mutation causes oligomeric aggregates that not evolve to fibrils or senile plaques but results in significant tau aggregation?
The article presents significant results to know the role played by mutations within the Aβ sequence and, therefore, may help to understand the amyloidogenic process in AD and its role in the pathophysiology and progression to dementia.
(Response) We have just published a review of our data on biochemical mechanisms involved in the development of FAD (Osaka) mutation (Tomiyama T, Shimada H. App Osaka mutation in familial Alzheimer's disease - Its discovery, phenotypes, and mechanism of recessive inheritance. Int J Mol Sci 2020; 21). The article is already available on line and will be issued in the special edition on amyloid beta of IJMS. If accepted, the current paper is proposed to be involved in the same edition. In that review paper, we summarized our data showing that Aβ oligomers alone can induce tau phosphorylation and neurodegeneration.
Reviewer 2 Report
The authors present a case report comparing the tau pathophysiology of proband FAD (Osaka) patient with SAD, advanced SAD and age-matched HCs. The report is quite interesting; however, this reviewer has following suggestions/concerns -
- Though interesting, the case study size for advanced SAD is 1 and it is difficult to compare the accumulation of tau between advanced SAD and proband FAD. This can be clearly seen with the scatter plots in figure 5 where both FAD and advanced SAD were in the range of SD for SAD. The reviewer understands that availability of subjects could be a major hindrance for including more patients in the study and therefore suggests that the authors can try to include at least one or two more advanced SAD patients. If not, this reviewer suggests that authors remove the some of the statements comparing SAD and FAD (lines 177 – 180)
- This reviewer recommends adding a sentence or two comparing the MRI scans of FAD to HC in figure 1
- More recent evidences have pointed the pertinent role of soluble oligomeric tau in AD onset and progression. The authors show that Osaka mutation results in heavy accumulation of tau NFTs but do not discuss the pathophysiology of tau oligomers in FAD(Osaka) pathology. This reviewer suggests that authors should discuss the occurrence, if any, of tau oligomers rather than NFTs in FAD(Osaka).
Minor points – Lines 70 – 76 – format the font size
Author Response
(Reviewer) The authors present a case report comparing the tau pathophysiology of proband FAD (Osaka) patient with SAD, advanced SAD and age-matched HCs. The report is quite interesting; however, this reviewer has following suggestions/concerns -
Though interesting, the case study size for advanced SAD is 1 and it is difficult to compare the accumulation of tau between advanced SAD and proband FAD. This can be clearly seen with the scatter plots in figure 5 where both FAD and advanced SAD were in the range of SD for SAD. The reviewer understands that availability of subjects could be a major hindrance for including more patients in the study and therefore suggests that the authors can try to include at least one or two more advanced SAD patients. If not, this reviewer suggests that authors remove the some of the statements comparing SAD and FAD (lines 177 – 180)
(Response) We agree to the reviewer’s comment that the number of the case of FAD and advanced SAD is limited. We withdrew our suggestion in the discussion that heavy tau accumulation in the lateral temporal lobe and parietal lobe is specific to FAD (Osaka) as below.
“Although the difference between FAD (Osaka) and advanced SAD was also noticeable in these ROIs, whether this distribution pattern is specific to FAD (Osaka) or just reflecting the advanced stage of AD in general requires further validation with more advanced SAD cases.”
(Reviewer) This reviewer recommends adding a sentence or two comparing the MRI scans of FAD to HC in figure 1
(Response) We appreciate the comment.
“These changes in FAD (Osaka) were all noticeable by comparing FAD (Osaka) to HC (Figure 1)”
(Reviewer) More recent evidences have pointed the pertinent role of soluble oligomeric tau in AD onset and progression. The authors show that Osaka mutation results in heavy accumulation of tau NFTs but do not discuss the pathophysiology of tau oligomers in FAD (Osaka) pathology. This reviewer suggests that authors should discuss the occurrence, if any, of tau oligomers rather than NFTs in FAD (Osaka).
(Response) Following the good suggestion by the reviewer #2, we discussed about the similar toxicity of tau oligomer as below.
“In addition to A beta, more recent evidences have pointed the pertinent role of soluble oligomeric tau in AD onset and progression. These oligomers may share a common mechanism of toxicity [17]. As the present case of FAD (Osaka) exhibited heavy tau burden on PBB3 PET, any other FAD mutation without tau accumulation if found might prove the toxicity of tau oligomer.”
(Reviewer) Minor points – Lines 70 – 76 – format the font size
(Response) We corrected the font size.